# CoTuning: A Large-Small Model Collaborating Distillation Framework for Better Model Generalization

## ABSTRACT

Model compression and distillation techniques have become essential for deploying deep learning models efficiently. However, existing methods often encounter challenges related to model generalization and scalability for harnessing the expertise of pre-trained large models. This paper introduces CoTuning, a novel framework designed to enhance the generalization ability of neural networks by leveraging collaborative learning between large and small models. CoTuning overcomes the limitations of traditional compression and distillation techniques by introducing strategies for knowledge exchange and simultaneous optimization. Our framework comprises an adapter-based co-tuning mechanism between cloud and edge models, a scale-shift projection for feature alignment, and a novel collaborative knowledge distillation mechanism for domain-agnostic tasks. Extensive experiments conducted on various benchmark datasets demonstrate the effectiveness of CoTuning in improving model generalization while maintaining computational efficiency and scalability. The proposed framework exhibits a significant advancement in model compression and distillation, with broad implications for research in the collaborative evolution of large-small models.

## CCS CONCEPTS

• **Computing methodologies** → **Image representations**.

## KEYWORDS

Knowledge Distillation, Collaborative Learning, Model Generalization, Model Compression

## 1 INTRODUCTION

In recent years, foundational models such as GPT [31] and Sora [28] have demonstrated strong generalization and versatility in various tasks and industries, bringing opportunities for the implementation of artificial intelligence. In practical applications, with a foundational model trained with generic data on the Cloud-device, we could then customize the cloud large model to a user-specific domain using tuning and model compression techniques according to different edge devices and application scenarios. Subsequently, through the evolutionary mode of collaborative evolution between edge and cloud, the effect of reducing energy consumption and improving overall model accuracy can be achieved. Among this,

*ACM MM, 2024, Melbourne, Australia*
© 2024 Copyright held by the owner/author(s). Publication rights licensed to ACM.
ACM ISBN 978-x-xxxx-xxxx-x/YY/MM
https://doi.org/10.1145/nnnnnnn.nnnnnnn

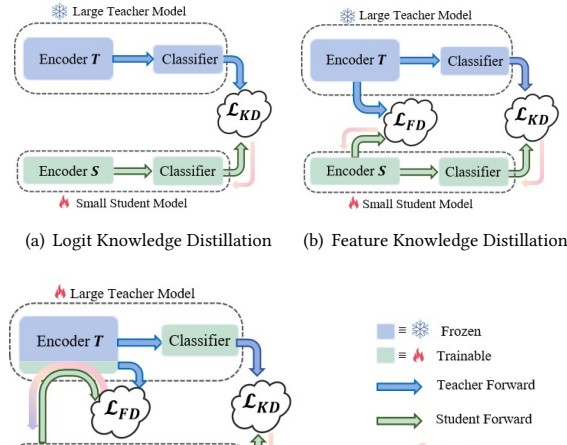

(a) Logit Knowledge Distillation  (b) Feature Knowledge Distillation

(c) Our Collaborating Knowledge Distillation

**Figure 1: Comparison of three kinds of distillation techniques. (a) and (b) distill the student model from a static teacher model, of which (a) updates the parameters according to class predictions while (b) relies more on the intermediate features. (c) Our Cotuning framework simultaneously updates the large-small models by conducting collaborative knowledge distillation between their intermediate layers.**

the effective transfer of global knowledge from large cloud-side model to small edge-side models is of great research significance.

Most existing methods [1, 18, 24, 36] focus on improving the precision of small edge models on specific tasks and scenes, but overlook the requirement for their generalization performance. Moreover, improving the generalization ability of edge-side small models is an effective means to reduce costs, mitigate data privacy issues, and facilitate collaborative evolution between edge and cloud models. This is because edge-side small models typically require cloud-side large models to undergo customized training based on downstream user data and task requirements. On one hand, the increase in the number of edge devices and application demands leads to a sharp rise in cloud-side training costs and edge-side storage space requirements. At the same time, adopting separate training for each edge can make it difficult for information exchange between multiple edges, posing challenges for subsequent model updates and evolution. Considering these factors, we propose training edge-side small models with good generalization capabilities.

A common solution is to use knowledge distillation techniques [6, 13, 40, 41] for model compression, which has played a crucial role in deploying and utilizing deep learning models efficiently. As shown in Figure 1(a) and 1(b), this specific approach involves using a static

 Anonymous Authors

pre-trained model to transfer structural knowledge [30, 33, 49] or feature distribution knowledge [21, 32, 48] to a smaller model. However, existing methods primarily focus on compressing knowledge from large-scale models into smaller counterparts to reduce computational and storage requirements. As a result, these techniques frequently encounter limitations due to the performance constraints of the upstream large models and tend to neglect considerations regarding model generalization.

To address these challenges, we propose a novel framework named CoTuning, aimed at enhancing model generalization performance while leveraging existing model compression and distillation techniques. Figure 1 depicts the comparison between the proposed method and traditional distillation approaches. Apparently, our CoTuning method builds upon the foundation of conventional knowledge distillation [13] but introduces strategies for collaborative learning [43] to mitigate the impact of upstream model performance limitations and effectively improve model generalization. This training strategy involves aligning the distribution of the smaller model with that of the larger model, while also incorporating collaborative optimization, allowing both models to iteratively enhance each other's performance. Technically, we explore a collaborative learning method that utilizes adapter-based tuning strategies [4, 52, 56] for large models of different scales. At the same time, large-small collaborative knowledge distillation is employed to ensure the flow and interaction of knowledge. It turns out that in many cases, collaborative learning between the cloud-edge models is beneficial for improving performance compared to traditional knowledge distillation methods.

Overall, CoTuning offers a straightforward yet powerful approach to enhance the generalization ability of neural networks. By harnessing collaborative learning and simultaneous optimization mechanisms, CoTuning overcomes the limitations of traditional knowledge distillation methods, leading to models that generalize better across diverse datasets and tasks. Extensive experiments conducted on various benchmark datasets, *i.e.,* cross-domain classification and retrieval tasks, demonstrate the efficacy of CoTuning in improving model generalization performance while maintaining computational efficiency and scalability. The proposed framework not only advances the field of model compression and distillation but also opens up new avenues for research in the collaborative evolution of large-small models.

The main contributions can be summarized as follows:

* **Adapter-based Co-tuning Framework between Cloud and Edge Model:** We propose a novel adapter-based co-tuning framework that facilitates collaborative learning between the cloud and edge model. This framework enables efficient knowledge transfer and adaptation from cloud-large models to edge-small ones, also leading to improved model generalization across distributed environments.
* **Collaborative Distillation Mechanism for Domain Agnostic Tasks:** We present a novel cloud-edge collaborative distillation mechanism tailored for domain-agnostic tasks, enabling the seamless transfer of knowledge between models trained on different datasets or domains. This mechanism enhances the adaptability and robustness of the CoTuning

framework, ensuring superior performance across diverse application scenarios.
* **Superior Experimental Results:** Our experimental findings demonstrate that the CoTuning framework achieves significant performance improvements across multiple benchmark datasets, showcasing its outstanding performance in terms of model generalization and efficiency.

## 2 RELATED WORK

### 2.1 Knowledge Distillation

In the past decades, knowledge distillation technique [9, 13, 20, 23, 25, 34] has been proven effective in transferring knowledge from larger, more capable teacher models to smaller, more suitable student models for practical applications across various domains. Common knowledge distillation methods include logit-based distillation, feature-based distillation, and relation-based distillation. For example, the logit-based Decoupled Knowledge Distillation(DKD) [54] method attempts to distill the knowledge by dividing the classical KD into target class knowledge distillation (TCKD) and non-target class knowledge distillation (NCKD). The DIST [17] method reveals that both the intra-class and the inter-class relations make positive impact on model distillation. These methods effectively transfer teacher knowledge to downstream models. However, on one hand, the performance of student model is constrained by the teacher model, and on the other hand, such distillation methods often result in student model with weak generalization.

### 2.2 Collaborative Learning

Collaborative Learning, where multiple models learn together and share insights, has proven to be an effective method for distilling knowledge across various tasks [5, 46, 47], such as classification [16, 44] and translation [51]. In comparison to distillation performed by a pre-trained static large network, collaborative learning among multiple models may somewhat achieve better performance. DML(Deep mutual learning) [53] utilizes a straightforward yet effective method to enhance the network's generalization capability by training collaboratively with a group of other networks. ML-LMCL(Mutual Learning and Large-Margin Contrastive Learning) [2] employs mutual learning to promote knowledge exchange between the model trained on clean manual transcripts and the model trained on ASR transcripts. These methods primarily focus on information exchange among models of equal scale. There is still significant research significance in exploring how to achieve collaborative training between models of different sizes and improve the models' generalization performance.

### 2.3 Parameter-Efficient Tuning

Efficient parameter tuning techniques [3, 8, 10, 27, 35, 42, 50] have become crucial for maximizing the utility of large pre-trained models in diverse domains such as natural language processing and computer vision. These techniques aim to minimize computational overhead while maintaining high performance levels. Two common approaches in this domain include prompted-based methods [10, 19, 22] and adapter-based methods [10, 14, 15]. During the fine-tuning of downstream tasks, these adapters, or soft prompts, are trained exclusively, while all pre-trained parameters remain

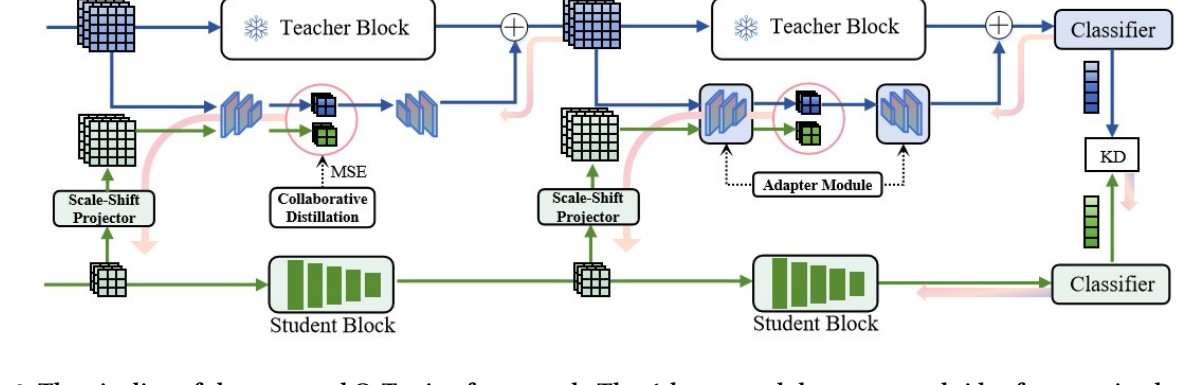

**Figure 2: The pipeline of the proposed CoTuning framework. The *Adapter* module serves as a bridge for cotuning between the large-small models. The *Scale-Shift Projector* maps features from the smaller model to a feature space with the same dimensions as the larger one. A *Collaborative Distillation* mechanism is performed to achieve simultaneous model updates.**

frozen. This ensures the generalization capability of the large model. The standard adapter-based module consists of a small neural network layer, typically comprising two fully connected layers and non-linear layer, such as RELU. The Scaled Parallel Adapter(SPA) builds upon this method by incorporating trainable low-rank matrices into transformer layers to mimic weight adjustments. This concept is an extension of LoRA's [15] principles, adapted specifically for adapters. In this work, we also exploit the scaled parallel adapter for the updates of the large model.

## 3 METHODOLOGY

In this section, we first detail some preliminary teacher-student distillation methods for better understanding. Second, we highlight the proposed Collaborative Knowledge Distillation framework to improve the model generalization with three pivotal parts. Finally, we describe how the CoTuning algorithm is optimized in an end-to-end fashion.

### 3.1 Preliminary

In the realm of model compression and knowledge distillation, the prevailing approach typically revolves around constraining the logits [13], or the middle-level features [12], of the student model to match those of the teacher model. This alignment is commonly achieved by minimizing the similarity measure between the predictions of the two models, *i.e.,* the Kullback-Leibler Divergence (KL) and the Mean Square Error (MSE).

Given a training set $\{(x_i, y_i)\}_{i=1}^{N}$ for teacher-student knowledge distillation, the model $\Phi(x) = \Phi_{\text{cls}}(f) \circ \Phi_{\text{fea}}(x)$ usually can be divided into two main parts for better feature distillation, *i.e.,* one feature extractor $\Phi_{\text{fea}}$ and a classifier $\Phi_{\text{cls}}$. Therefore, the feature vector can be calculated by $f_i = \Phi_{\text{fea}}(x_i)$, while the logit output can be gained from $\boldsymbol{p}_i = \Phi_{\text{cls}}(f_i)$. Then, the common CE loss for model training can be formulated as:

$$L_{\text{CE}} = -\frac{1}{N} \sum_{i=1}^{N} \sum_{j=1}^{K} y_{i,j} \log \hat{p}_{i,j}, \qquad (1)$$

where $\hat{p}_{i,j} = \exp(p_{i,j})/\sum_j \exp(p_{i,j})$ means normalized probability, and K is the number of categories for classification.

As most pertinent literatures [13, 23, 54] introduced, the vanilla knowledge distillation is always constrained with the KL loss, which can be formulated as:

$$L_{\text{KD}} = \frac{1}{N} \sum_{i=1}^{N} L_{\text{KL}}(\hat{\boldsymbol{p}}_i^t || \hat{\boldsymbol{p}}_i^s), \qquad (2)$$

where $L_{\text{KL}}(\hat{\boldsymbol{p}}_i^t || \hat{\boldsymbol{p}}_i^s) = \sum_{j \in K} \hat{p}_{i,j}^t \ln(\hat{p}_{i,j}^t / \hat{p}_{i,j}^s)$, and $t/s$ mean logits from teacher or student model, respectively.

When focusing on feature distillation [48], the MSE loss is often the preferred choice to constrain the alignment between features. This can be formulated as follows:

$$L_{\text{MSE}} = \frac{1}{N} \sum_{i=1}^{N} \|f_i^t - f_i^s\|_2^2. \qquad (3)$$

As advanced methods, the RKD [33] and the DIST [17] considered match the teacher model and the student one with the relations, containing inter-relations between different instances in one batch and intra-relations among each category.

### 3.2 Collaborative Knowledge Distillation

In this paper, we primarily introduce a mechanism for collaborative learning involving a cloud-side fundamental model and an edge-side small model. The objective is to attain comparable feature distributions across both models, aiming to enhance the small model's generalization performance across various scenarios for improved deployment in practical applications. Specifically, regarding the fundamental model, fine-tuning all parameters often triggers overfitting. Hence, we employ a parameter-efficient tuning technique called scaled parallel adapter learning to adjust its model parameters. Conversely, for the small-sized models on the edge side, we optimize them directly using a full parameter training mode. Additionally, we develop a feature projection module to align their feature distributions with those of the fundamental model, of which

is optimized alongside the scaled adapter in a step-by-step manner to ensure stable performance. As shown in Figure 2, our method consists of three parts as follows: adapter-based co-tuning module, scaled-shift feature projection module, and novel knowledge collaborative distillation mechanism.

*3.2.1  Adapter-based Co-tuning Module.* Instead of fine-tuning all parameters of the fundamental model, we introduce adapter layers that adaptively adjust the model's representations to match those of the edge models. Specifically, when employing adapters for fine-tuning to adapt to downstream tasks, the adapter adjusts the model's output offset on specific tasks by first reducing and then increasing the dimensions of the fundamental model's feature representation. Throughout this process, the feature distribution space becomes more condensed after the adapter reduces the dimensionality, resulting in lower feature dimensions while still preserving critical features in the data. This helps in reducing redundant information and improving computational efficiency while accurately capturing task-relevant key features. In the subsequent expansion of feature dimensions, the adapter remaps the low-dimensional features back to the original feature dimension space of the fundamental model. This process enhances the expressiveness of the features and makes them more suitable for specific tasks. By adopting this approach, the adapter effectively fine-tunes the model's feature representation, mitigating the risks of over-fitting and enhancing the model's performance and generalization capabilities.

Considering the reasons mentioned above, our approach seeks to leverage the adapter's capabilities to acquire compact feature representations for downstream tasks and promote collaborative learning between the fundamental model and the small-sized models at the edge. To achieve this, we utilize a scaled parallel adapter $\theta_k$ for each block of the fundamental model, where $\theta_k = \{\theta_k^d, \theta_k^r, \theta_k^u, \theta_k^s\}$. Here $\theta_k^d$ indicates down-sampling layer, $\theta_k^r$ is non-linear layer, $\theta_k^u$ is up-sampling layer, $\theta_k^s$ represents a scaling factor, the subscript $k$ indicates which layer the adapter module is employed. For any input feature $f$, the output features $\overline{f}$ through adapter $\theta_k$ can be represented as follows,

$$\overline{f}_k = \theta_k(f) = \theta_k^u(\theta_k^r(\theta_k^d(f))) * \theta_k^s, \tag{4}$$

where $k$ indicates which adapter module is exploited for feature extraction.

*3.2.2  Scaled-Shift Feature Projection Module.* The scaled-shift feature projection module aims to align the feature distributions of the edge models with those of the fundamental model. This is achieved by projecting the feature representations of the edge models into a common feature space, where they can be compared and aligned with the representations of the fundamental model. To ensure stability and robustness, we introduce a scaled-shift mechanism that adjusts the projection parameters in a controlled manner.

Figure 3 elucidates the precise operations of this module. In particular, for a specific intermediate layer $k$ of the small model, whose features defined as $f_k^s$, we initially employ a standard projection module to map it to a space with the identical dimensionality as the features of the large model. Generally speaking, the parameters of the projection module at the $k$ layer is concluded

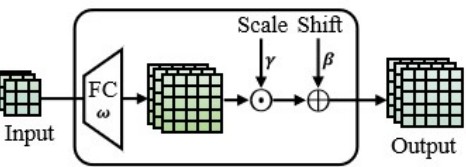

**Figure 3: Pipeline of the scaled-shift projection module. The input features are the low-dimensional features from the intermediate layer of the student model. After linearly projecting them into a high-dimensional space, they undergo feature-wise scaling and shifting operations to obtain the final output features. These features are then utilized for subsequent interactions between the large-small models.**

as $\phi_k = \{\omega_k, \beta_k, \gamma_k\}$, where $\omega_k, \gamma_k$ and $\beta_k$ corresponds to the linear projection layer, the scaling and the shifting parameters. First, we utilize $\omega_k$ to project the student features into the same dimension as the teacher model, then the features can be represented as $f_k^{s'} = \omega_k(f_k^s)$. Subsequently, we further utilize feature-wise scaling and shifting operations to modify the significance and offset of each input feature, while retaining the inherent physical interpretation of the features intact. Therefore, the feature output $f_k^{s,\phi}$ after passing through this projection module can be derived as follows,

$$f_k^{s,\phi} = \gamma_k \odot \omega_k(f_k^s) + \beta_k. \tag{5}$$

During the parameter initialization phase, note that the scaling and shifting parameters of this module are set to 1 and 0, respectively.

*3.2.3  Novel Collaborative Knowledge Distillation Mechanism.* Traditional feature distillation methods typically entail calculating the correlation between the intermediate layer features of the student and teacher models, followed by optimizing the student model's feature representation through correlation constraints to align it with that of the teacher model. Therefore, the key issue lies in how to define the consistency between the distributions of two feature representations. Common approaches involve using similarity metrics such as L2 loss, similarity preserving loss, and feature structural loss to optimize the alignment between feature distributions. However, in the FCFD [25] method, it is emphasized that the similarity between features is not solely dictated by the features themselves but is rather defined by how subsequent layers will interpret, decode, and manipulate them. This insight inspires us to pursue a more seamless integration of the projection and adapter modules, rather than merely aligning their outputs at the corresponding intermediate layers. More specifically, we believe that if the features of the teacher model and the student model have consistent representations, then a teacher-friendly adapter structure should produce similar effects on the student model.

For a certain intermediate layer $k$, we could obtain the projected student feature $f_k^{s,\phi}$ and the corresponding teacher feature $f_k^t$. Then, with the next layer adapter module denoted as $\theta_{k+1}$, we employ it to propagate these features forward and acquire the corresponding outputs,

$$\overline{f}_{k+1}^{s,\phi}, \overline{f}_{k+1}^t = \theta_{k+1}(f_k^{s,\phi}), \theta_{k+1}(f_k^t). \tag{6}$$

Thus a straightforward form of feature distillation can be achieved using the following appearance loss,

$$L_{\text{MSE}} = \|\overline{f}_{k+1}^{s,\phi} - \overline{f}_{k+1}^{t}\|_2^2. \tag{7}$$

As previously discussed, considering the dimensionality reduction operation in the adapter, it can project the original representation into a low-dimensional compact feature expression space. In this case, we can conduct knowledge distillation in this reduced-dimensional feature space. In other words, we exclusively employ module $\theta_{k+1}^d$ to conduct the feature alignment operations. Then the collaborative feature distillation loss can be calculated as,

$$L_{\text{MSE}}^{t\&s} = \|\theta_{k+1}^d(f_k^{s,\phi}) - \theta_{k+1}^d(f_k^t)\|_2^2. \tag{8}$$

Compared to directly aligning the features after two mappings, associating them at the intermediate layer offers more advantages. Firstly, reduced-dimensional features can partially eliminate redundancy and better capture essential data information. Constraining features in this space mitigates model overfitting and enhances generalization. Moreover, conducting computations in low-dimensional feature space often lowers computational resource usage. In conclusion, this novel collaborative knowledge distillation mechanism further enhances the adaptability of the edge models by incorporating domain-specific knowledge into the training process. By leveraging domain-specific information, such as task-related features or contextual cues, we enable the edge models to better generalize to specific application scenarios. This mechanism is integrated into the training pipeline to ensure that the edge models effectively capture and adapt to relevant knowledge during training.

## 3.3 Model Training

The optimization and model training process involve fine-tuning the fundamental model using scaled parallel adapter learning and training the edge models using full parameter training. The feature projection module and the novel knowledge distillation mechanism are optimized alongside the scaled adapter in an end-to-end fashion to ensure consistent and stable performance across all components of the framework. The total loss for our method is formulated as:

$$L_{\text{CoTuning}} = L_{\text{CE}}^t + L_{\text{CE}}^s + \alpha L_{\text{KD}}^s + \lambda \sum_{k \in K} L_{\text{MSE}}^{t\&s}, \tag{9}$$

where $K$ is the number of blocks with co-tuning module, $\alpha$ and $\lambda$ are the hyper-parameters for controlling the influence of the logit-based and the feature-based distillation losses, respectively. Notice that here we utilize the same formulation of DKD [54] for $L_{\text{KD}}^s$. The whole training processing with CoTuning framework is summarized in Algorithm 1.

## 4 EXPERIMENTS

### 4.1 Datasets and Evaluation Metrics

To validate the effectiveness of the proposed method, we conducted experiments on both classification and retrieval tasks. For the classification task, we utilized the OfficeHome [39] benchmark. In Office-Home, there are 4 domains of images with a total of 65 classes, including Art, Clipart, Product, and Realworld. We train the model on each domain and do cross-domain testing on the others. For the retrieval task, we conducted training on the VIPeR [7] and Market [55]

---

**Algorithm 1:** CoTuning

**Input**: Training dataset $D$ ; Training Epochs $Ep$; teacher model $\Phi^t$; student model $\Phi^s$; adapters $\theta$; projectors $\phi$; selected layers $K$ for feature distillation

**Output**: student model $\Phi^s$

**while** $epoch \leqslant Ep$ **do**

   **for** *data batch* $x$ *in* $D$ **do**

      **Forward propagation**, obtain class prediction and intermediate features for $k \in K$ layers:

         $p^s, f_k^s = \Phi^s(x)$

         $f_k^{s,\phi} = \phi_k(f_k^s)$

         $p^t, f_k^t = \Phi^t(x)$

         $\overline{f}_{k+1}^{s,\phi,d}, \overline{f}_{k+1}^{t,d} = \theta_{k+1}^d(f_k^{s,\phi}), \theta_{k+1}^d(f_k^t)$

      **Calculate the Loss**,

         Calculate $L_{\text{CE}}^t, L_{\text{CE}}^s$ according to Eq 1 with $p^t, p^s$

         Calculate $L_{\text{KD}}^s$ according to [54] with $p^t, p^s$

         Calculate $L_{\text{MSE}}^{t\&s} := \|\overline{f}_{k+1}^{s,\phi,d} - \overline{f}_{k+1}^{t,d}\|_2^2$

      **Backward propagation**,

         updates $\Phi^s, \phi, \theta$ simultaneously

   **end**

**end**

---

datasets, and testing on the VIPeR, Market, CUHK-SYSU(SYSU) [45], VeRI [26] and Inshop-Clothes(Inshop) [29] datasets. The VIPeR dataset contains 632 classes with a total of 1264 images, while the Market dataset contains 1501 classes with a total of 32,668 images. Both of these two datasets are fine-grained retrieval datasets focused on pedestrians. SYSU is also a pedestrian retrieval dataset, VeRI is a vehicle retrieval dataset, and Inshop is a retrieval dataset for fashion products.

For evaluation, we employ the classification Accuracy(ACC) and Rank-1 accuracy (R-1) for the classification and retrieval task. We further report the average performance for each task.

### 4.2 Implemented Details

We use DeiT-Base/16 [37] pre-trained on Imagenet as our backbone and keep it frozen in the entire training stage, and the inner dimension of the adapter module is set as 32. We appended trainable adapter modules to the last 6 layers of the pre-trained model. The amount of trainable parameters is around 0.6M, which accounts for 0.69% of the total parameters of the pre-training model. The reported student model exploited the DeiT-Tiny/16 structure. For data processing, all images are resized to $224 \times 224$ for all datasets, and data augmentation involves random crop and random erasing. For optimizing, we set the batch size as 128, and use Adam for optimization which trains 300 epochs for each task. The learning rate is initialized as $3.5 \times 10^{-4}$, which is then decreased via the Cosine Annearling strategy. All the balanced factors for losses are set to 1 and the temperature $T$ for knowledge distillation is set to 4.

### 4.3 Compared Methods

The methods compared in this work conclude:

**Table 1: Comparison of the classification task on the Office-Home dataset. The reported result is trained on one source domain and then test on the others. $\bar{s}_{ACC}$ indicates the average cross-domain classification accuracy.**

| Source | Target | | | Avg |
|---|---|---|---|---|
| Arts → | Product | Clipart | Realworld | $\bar{s}_{ACC}$ |
| SPA | 76.5 | 58.4 | 82.4 | 72.4 |
| VKD | 47.9 | 33.1 | 58.6 | 46.5 |
| SKD | 53.6 | 39.3 | 64.0 | 52.3 |
| DKD | 53.9 | 39.9 | 64.4 | 52.7 |
| DIST | 47.7 | 36.7 | 58.9 | 47.8 |
| DML | 45.0 | 32.2 | 55.2 | 44.1 |
| RKD | 48.9 | 34.8 | 59.3 | 47.7 |
| SP | 42.5 | 31.5 | 54.6 | 42.9 |
| PKT | 48.8 | 35.0 | 60.9 | 48.2 |
| Scratch | 39.3 | 29.5 | 51.6 | 40.1 |
| Ours | 56.2 | 40.4 | 66.2 | 54.3 |
| Product → | Arts | Clipart | Realworld | $\bar{s}_{ACC}$ |
| SPA | 65.6 | 81.9 | 51.6 | 66.4 |
| VKD | 21.4 | 48.7 | 27.7 | 32.6 |
| SKD | 28.2 | 56.2 | 34.7 | 39.7 |
| DKD | 26.8 | 55.0 | 33.9 | 38.6 |
| DIST | 26.0 | 53.1 | 30.0 | 36.4 |
| DML | 23.1 | 49.7 | 29.6 | 34.1 |
| RKD | 21.4 | 47.9 | 27.9 | 32.4 |
| SP | 20.0 | 45.9 | 27.0 | 31.4 |
| PKT | 21.5 | 49.4 | 28.4 | 33.1 |
| Scratch | 17.8 | 42.4 | 23.2 | 27.8 |
| Ours | 31.6 | 60.0 | 37.5 | 43.0 |
| Clipart → | Arts | Product | Realworld | $\bar{s}_{ACC}$ |
| SPA | 68.1 | 76.4 | 79.8 | 74.8 |
| VKD | 27.1 | 46.1 | 43.8 | 39.0 |
| SKD | 33.2 | 51.4 | 50.3 | 45.0 |
| DKD | 33.4 | 50.8 | 49.6 | 44.6 |
| DIST | 29.2 | 46.0 | 45.3 | 40.2 |
| DML | 31.1 | 48.2 | 47.8 | 42.4 |
| RKD | 27.5 | 45.4 | 43.0 | 38.6 |
| SP | 26.6 | 43.4 | 41.4 | 37.1 |
| PKT | 29.7 | 46.4 | 44.9 | 40.3 |
| Scratch | 20.6 | 39.0 | 37.0 | 32.2 |
| Ours | 36.6 | 55.2 | 54.4 | 48.7 |
| Realworld → | Arts | Product | Clipart | $\bar{s}_{ACC}$ |
| SPA | 73.0 | 82.4 | 54.6 | 70.0 |
| VKD | 42.1 | 63.4 | 36.6 | 47.4 |
| SKD | 48.9 | 69.7 | 41.2 | 53.3 |
| DKD | 47.8 | 67.7 | 40.9 | 52.1 |
| DIST | 44.8 | 64.9 | 38.8 | 49.5 |
| DML | 44.4 | 65.0 | 38.1 | 49.2 |
| RKD | 42.0 | 63.7 | 37.0 | 47.6 |
| SP | 39.8 | 61.6 | 31.5 | 43.1 |
| PKT | 36.2 | 58.1 | 33.4 | 42.6 |
| Scratch | 36.0 | 58.8 | 31.5 | 42.1 |
| Ours | 49.8 | 71.2 | 43.8 | 54.9 |

1) Scaled Parallel Adapter(SPA) [10]: it attaches a small number of parameters to the fundamental model and efficiently fine-tunes them. This method can be considered as the upper bound.

2) Vanilla Knowledge Distillation(VKD) [13]: it mimicks the difference between teacher and student predictions via the Kullback-Leibler (KL) divergence.

3) Spherical Knowledge Distillation(SKD) [9]: it normalizes the predictions of both the teacher and the student models according to the magnitude confidence.

4) Decoupled Knowledge Distillation(DKD) [54]: it divides the classical KD into target class knowledge distillation (TCKD) and non-target class knowledge distillation (NCKD).

5) Distillation from A Stronger Teacher(DIST) [17]: it executes knowledge distillation with both inter-relations and intra-relations.

6) Deep Mutual Learning(DML) [53]: it trains multiple students that enable learn collaboratively throughout the training process.

7) Relational Knowledge Distillation(RKD) [33]: it proposes to distill complex relationships and dependencies between feature representations from the teacher and student model.

8) Similarity-Preserving Knowledge Distillation(SP) [38]: it employs pairwise activation similarities for distillation.

9) Probabilistic Knowledge Transfer(PKT) [34]: it minimizes the divergence between the probability distribution between the teacher and student models.

10) Training from scratch(Scratch): it trains models without leveraging pre-trained knowledge or parameters.

## 5 RESULTS

### 5.1 Comparisons to state-of-art approaches

Table 1 and Table 2 respectively showcase the performance of the model in classification and retrieval tasks. Among all reported methods, SPA represents the results obtained by fine-tuning based on the teacher model, and its performance can be regarded as the upper limit of the distilled student model's performance. For the classification task, the proposed method achieves an average accuracy of 54.3%, 43.0%, 48.7%, and 54.9% across the Arts, Product, Clipart, and Realworld domains, respectively. It outperforms the second-best distillation method by an average generalization performance improvement of 1.6% to 3.7%. For the retrieval task, the proposed method achieves an average accuracy of 45% and 57.8% for the VIPeR and the Market dataset. On the Market dataset, it outperforms the second-best DIST method by an average performance improvement of 2.2%. On the VIPeR dataset, it exceeds the average performance of the second-best method by 3.5%. Through comparisons, we observe that most existing logit-based distillation methods perform well when tailored for specific tasks. However, once these methods are applied to cross-domain scenarios or tasks, the performance of the distilled student appears significantly decreases. This could be attributed to the fact that, in comparison to logit-based distillation, our feature-based distillation method typically allows for the preservation of more detailed information, consequently leading to better generalization capabilities.

### 5.2 Comparisons with different student models

In this section, we evaluate the generalization performance with different student models. We continue to use DeiT-B as the pre-trained

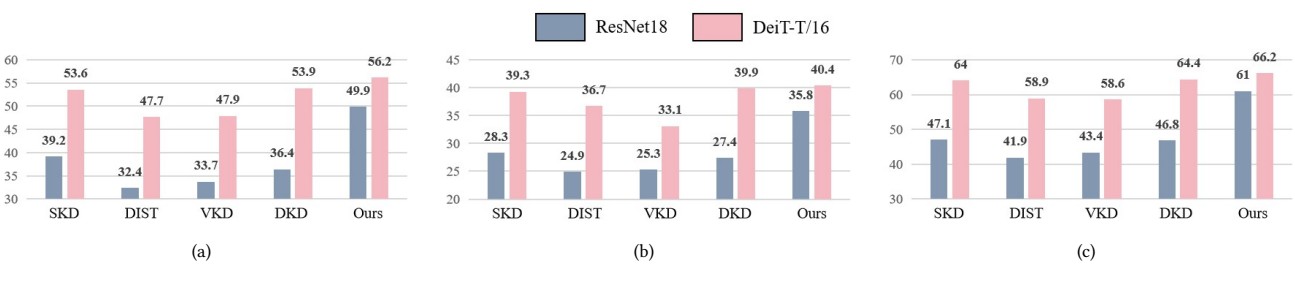

**Figure 4: Comparison with different student models on the OfficeHome datasets. We train the student model using the Arts split data, and show the performance on the (a)Product, (b)Clipart, and (c)RealWorld domain. The gray and pink bar corresponds to the classification accuracy of the Resnet18 and the DeiT-T/16 model.**

**Table 2: Comparison of the retrieval task on the VIPeR and the Market dataset. We report the Rank-1 matching rate in the table. $\bar{s}_{R_1}$ indicates the average retrieval Rank-1 score.**

| Source | Target | | | | | Avg |
|---|---|---|---|---|---|---|
| Market→ | VIPeR | Market | SYSU | VeRI | InShop | $\bar{s}_{R_1}$ |
| SPA | 36.4 | 88.5 | 80.1 | 47.9 | 56.4 | 61.9 |
| VKD | 28.2 | 89.0 | 70.8 | 31.5 | 43.3 | 52.6 |
| SKD | 32.9 | 90.5 | 72.9 | 35.7 | 43.5 | 55.1 |
| DKD | 31.0 | 90.9 | 73.9 | 34.4 | 45.6 | 55.2 |
| DIST | 32.6 | 90.4 | 74.3 | 35.9 | 44.4 | 55.6 |
| DML | 28.8 | 90.0 | 71.7 | 34.9 | 43.8 | 53.8 |
| RKD | 28.2 | 89.1 | 71.7 | 32.1 | 44.4 | 53.1 |
| SP | 25.3 | 86.9 | 66.8 | 29.0 | 42.0 | 50.0 |
| PKT | 31.3 | 90.0 | 70.5 | 32.6 | 44.2 | 53.7 |
| Scratch | 22.5 | 83.8 | 61.9 | 26.3 | 40.0 | 46.9 |
| Ours | 34.2 | 90.8 | 77.2 | 39.7 | 47.3 | 57.8 |
| VIPeR → | VIPeR | Market | SYSU | VeRI | InShop | $\bar{s}_{R_1}$ |
| SPA | 54.7 | 41.2 | 69.0 | 43.2 | 49.1 | 51.4 |
| VKD | 50.6 | 23.3 | 53.1 | 27.8 | 34.3 | 37.8 |
| SKD | 52.8 | 27.1 | 53.8 | 28.4 | 33.0 | 39.0 |
| DKD | 57.6 | 26.8 | 56.3 | 26.6 | 34.6 | 40.3 |
| DIST | 55.7 | 27.6 | 56.3 | 29.9 | 34.9 | 40.9 |
| DML | 56.0 | 23.9 | 54.4 | 25.6 | 33.7 | 38.7 |
| RKD | 57.3 | 27.0 | 58.9 | 31.0 | 33.4 | 41.5 |
| SP | 47.8 | 20.3 | 51.2 | 25.0 | 32.4 | 35.3 |
| PKT | 56.3 | 26.2 | 55.3 | 28.5 | 33.8 | 40.0 |
| Scratch | 47.5 | 23.7 | 52.2 | 24.4 | 34.8 | 36.5 |
| Ours | 59.8 | 32.6 | 61.9 | 34.1 | 36.6 | 45.0 |

large model but select small models with CNN and transformer architectures, validating them on the OfficeHome classification task. We choose ResNet-18 [11] and Deit-Tiny as the small model for comparison. Here, we use the OfficeHome-Arts dataset for training and evaluate the generalization performance of the student models in the remained three domains: Product, Clipart, and Real World. In Figure 4, we show the classification accuracy of SKD, DKD, DIST, VKD, and the proposed method. The three subplots (a),(b) and (c) respectively depict the classification accuracy in the

Product, Clipart, and Real-world scenarios. The gray and pink bars correspond to the results obtained using ResNet18 and Deit-T/16 structures as the student models, respectively.

It can be seen that when both the teacher and student models are of Transformer architecture, the distilled student models tend to achieve better results. Our method demonstrates a 3% improvement in average performance compared to the second-best DKD method. However, when the teacher and student models adopt two different structures, Transformer and CNN respectively, the performance of the distilled student models significantly declines. Specifically, the average performance of DKD decreases from 52.6% to 36.9%, and SKD decreases from 52.5% to 38.2%. In comparison, we observe that our method still achieves a favorable average classification accuracy of 48.9% in the scenario of CNN student models. We speculate that this may result from the beneficial effects of collaborative feature learning between the adapter and the projection model. In traditional distillation methods, the learning of the feature projection module and the teacher model often follows an independent/isolated learning strategy. This could result in a gap between the features of the student and teacher models, particularly when substantial structural differences exist between the two. We hypothesize that achieving improved feature alignment requires a certain level of correlation between the projection modules of the student and teacher models. In this work, we employ feature-level collaborative learning to align the features and achieve this effect. The results of the experiments to some extent validate our speculation.

## 5.3 Validation of feature collaborative knowledge distillation(FCKD)

In this section, our aim is to analyze the impact of feature collaborative distillation involved in this method on model generalization. We conduct validation on the OfficeHome dataset. Specifically, we demonstrate the performance of models under six different training strategies, as illustrated in Figure 5. Among them, (a) and (b) both employ training with static pre-trained parameters, with (b) additionally incorporating distillation loss based on feature mappings compared to (a). (c) and (d) represent training modes using the concept of mutual learning that both the teacher and the student model require updates. Here (c) involves only distillation loss at the logits level, while (d) adds distillation loss based on standard

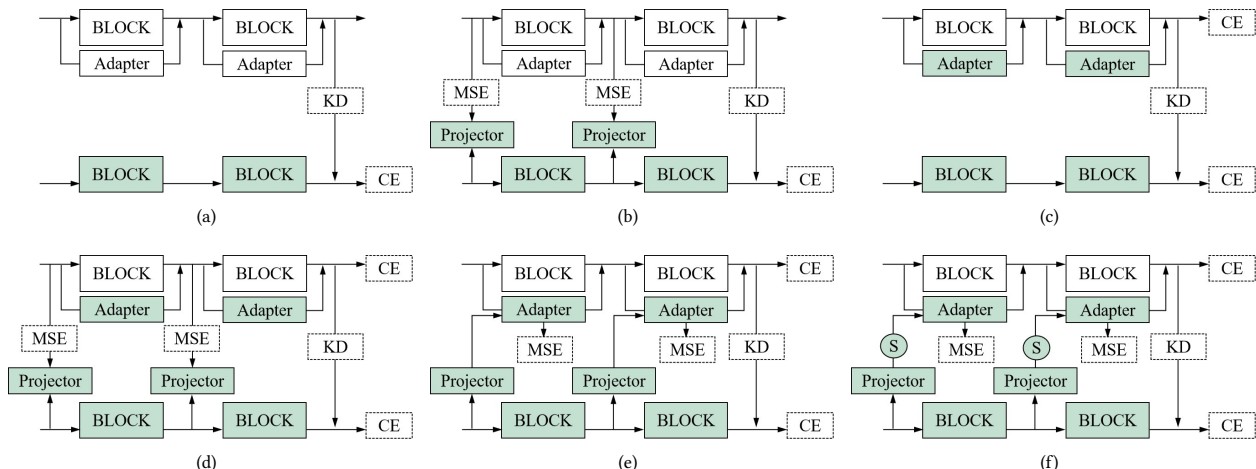

**Figure 5: Illustration of six training strategies. (a) static teacher with logit-KD loss. (b) static teacher with logit-KD loss and feature-level MSE loss. (c) trainable teacher with logit-KD loss. (d) trainable teacher with logit-KD loss and feature-level MSE loss. (e) our feature collaborating learning method. (f) our scale-shift feature collaborating learning method. Modules in green indicate that they are trainable.**

| Method | Setting | | | Distillation Loss | | Source: Arts | | | |
|--------|---------|-------------|------|---------|-------------|---------|---------|-----------|------|
|        | Teacher | Scale-Shift | FCKD | Logit-KD | Feature-MSE | Product | Clipart | Realworld | Avg |
| (a) | static | ✗ | ✗ | ✓ | ✗ | 53.9 | 39.9 | 64.4 | 52.7 |
| (b) | static | ✗ | ✗ | ✓ | ✓ | 53.2 | 40.4 | 64.4 | 52.7 |
| (c) | trainable | ✗ | ✗ | ✓ | ✗ | 50.0 | 35.3 | 60.9 | 48.7 |
| (d) | trainable | ✗ | ✗ | ✓ | ✓ | 55.5 | 32.3 | 45.3 | 44.5 |
| (e) | trainable | ✗ | ✓ | ✓ | ✓ | 55.6 | 40.1 | 65.9 | 53.9 |
| (f) | trainable | ✓ | ✓ | ✓ | ✓ | 56.2 | 40.4 | 66.2 | 54.3 |

**Table 3: Comparison with different training settings. (a) and (b) indicate standard distillation strategies without or with the feature MSE loss. (c) and (d) indicate mutual distillation of teacher and student models without or with the feature MSE loss. (e) and (f) correspond to our proposed collaborating distillation method without or with a scale-shift operation.**

feature projection on top of that. (e) and (f) adopt the distillation strategy proposed in this work, with (e) removing the scaling factor from the feature projection module. The comparative results are shown in Table 3.

According to the performance comparison in Tables 3 (a) and (b), we observe that with a static pre-trained teacher model, directly aligning features does not lead to effective performance improvement. The results in Table 3(c) and (d) indicate that in the mutual learning distillation strategy, where both the student and teacher models are optimized simultaneously, directly aligning features can even result in a significant performance decrease. The result in Table 3(e) confirms that exploiting our collaborative feature learning between the projection and the adapter module can lead to a 5.2% increase in average performance. It precisely validates the positive impact of jointly optimizing features through adapter and projector modules on enhancing model generalization performance. In addition to this, the result of (f) shows that adding a scaling factor can further enhance our generalization performance by 0.4%.

It demonstrates that adding the operation of scaling and shift also contributes to the improvement of model generalization.

## 6 CONCLUSION

Existing model compression techniques often face limitations due to the performance constraints of the upstream large models and tend to overlook concerns related to the generalization of small models. Our goal is to train small models with good generalization capabilities. In contrast to existing methods that mostly extract knowledge from static teacher models or simply align feature representations using projection modules, we propose collaboratively distilling knowledge between large and small models. Through the collaborative action of adapters and projection modules, we conduct feature knowledge interaction in a low-dimensional compact representation space similar to that of the teacher model. Extensive experiments validate that models trained in this way reveal good generalization performance across multiple tasks and scenarios.

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
