# OpenReview forum: "CoTuning: A Large-Small Model Collaborating Distillation Framework for Better Model Generalization"
_acmmm.org/ACMMM/2024/Conference — MM2024 Poster_

### Official Review · Reviewer_MBjv · 2024-05-23

**Rating:** 4
**Confidence:** 3

**Summary:**

This paper introduces CoTuning, a novel framework designed to enhance the generalization ability of neural networks by leveraging collaborative learning between large and small models. CoTuning overcomes the limitations of traditional compression and distillation techniques by introducing strategies for knowledge exchange and simultaneous optimization.

**Strengths:**

The paper presents innovative concepts, particularly with the introduction of the adapter structure, which provides significant insights into traditional feature-based distillation methods. And the illustrations are excellent, clearly depicting the algorithm's workflow and details, making it easy to understand. Moreover, experiments are comprehensive, offering thorough comparisons and detailed explanations of various baseline methods.

**Limitations:**

Generalization and Scalability:

The generalization and scalability of the proposed method may be limited due to the necessity of incorporating adapters in the teacher model. It may cause two problems as follow:
On one hand, High-performance teacher models, such as GPT and Sora, often do not allow external modifications to their intermediate layers. Therefore, the proposed approach may not be feasible in scenarios where the teacher model's feature maps cannot be externally modified.
On the other hand, The requirement for the adapters implies that for each different downstream task, a specific adapter must be allocated to the teacher model. As the number of downstream devices increases, the proliferation of adapters could lead to redundancy in the teacher model and increase communication costs between cloud and edge devices. Thus, a quantitative analysis of the scalability and cost of the proposed method is recommended to address these issues.

The recommendations for the two points above are:  1)Consider methods that do not require altering the teacher's feature map to enhance generalization further. 2) Further analyze the impact of an increasing number of edge devices on the computational and storage pressure on the cloud device to demonstrate scalability.


Clarity and Conciseness:

The writing can be made more concise. Some sentences are overly long and complex, which may obscure key points. For example, the sentence in the Introduction section, "As a result, these techniques frequently encounter limitations due to the performance constraints of the upstream large models and tend to neglect considerations regarding model generalization," could be simplified for better readability.
The explanation of key concepts should be clearer and more direct. Simplifying the language and streamlining the narrative would enhance the paper's readability.
Consistency in Terminology:

The term "generalization" is used inconsistently throughout the paper. It is crucial to specify whether it refers to the teacher model, the student model, or the entire algorithm.
For instance, the Abstract mentions challenges related to " ... model generalization and scalability for harnessing the expertise of pre-trained large models", while the Introduction discusses " ... improving the generalization ability of edge-side small models.". Elsewhere in the text, there are several instances of "large model generalisation", "small model generalisation", "model generalisation", "neural networks generalisation", "fundamental model overfitting", and other descriptions related to "generalisation", which may have inconsistent implications.
A unified and clear definition of "generalization" should be provided to avoid confusion. It may be necessary to avoid inconsistencies in the use of the term and ensure that its meaning is clear in contexts where it is used.

**Suitability:**

2

---

### Official Review · Reviewer_WKH6 · 2024-06-03

**Rating:** 3
**Confidence:** 3

**Summary:**

The paper introduces CoTuning, a novel framework designed to enhance the generalization ability of neural networks by leveraging collaborative learning between large and small models. CoTuning overcomes limitations of traditional compression and distillation techniques by introducing strategies for knowledge exchange and simultaneous optimization. The framework includes an adapter-based co-tuning mechanism between cloud and edge models, a scale-shift projection for feature alignment, and a collaborative knowledge distillation mechanism for domain-agnostic tasks. Experiments on benchmark datasets demonstrate the effectiveness of CoTuning.

**Strengths:**

1. The method presented in this paper is straightforward, and the explanation is clear and easy to understand.

2. The experiments in this paper compare a large number of baselines.

**Limitations:**

1. The paper aims to enhance the domain generalization capability of distillation models but does not specifically explain how the current method achieves this. The current method could serve as a knowledge distillation technique for more general classification tasks.

2. Since the proposed method requires the gradients of the teacher model, it may present efficiency issues. Could the author compare the memory consumption and training time differences between the current method and naive knowledge distillation?

3. The method requires adding extra features to the teacher model's features. Will these additional features introduced during the forward pass of the trained teacher model affect its accuracy? Could the authors report the teacher model's accuracy during the training process?

**Suitability:**

2

---

### Official Review · Reviewer_M3co · 2024-06-04

**Rating:** 3
**Confidence:** 3

**Summary:**

This paper proposes a knowledge distillation solution used in model tuning. Specifically, a teacher network and a student network are tuned together. Only some adapter layers in the teacher are trainable. Intermediate features in the student network after a projector and a scale-shift operation are expected to be consistent with those in teacher's adapters. Experiments on some classification and retrieval datasets demonstrate that the proposed method outperforms other widely adopted distillation methods.

**Strengths:**

1. The performance outperforms vanilla knowledge distillation schemes by a relatively large margin on the adopted benchmarks.
2. The writing is logical and coherent.

**Limitations:**

1. The setting is a combination of fine-tuning and online knowledge distillation, which may not be too much novel.
2. It is unclear whether the student network is pre-trained. If yes, this will be a very strong assumption, and the setting would be degraded into an online knowledge distillation problem. If no, the performance of fine-tuning the pre-trained student is missing as a baseline. I will assume the student network is not pre-trained in the following comments.
3. Although the proposed method improves the performance, I am afraid that it does not address the problem substantially, given the gap of pre-training. If the pre-trained dataset is (partially) available, distilling on the pre-trained dataset firstly, or adding some regularization related with the pre-trained data, should be a more suitable choice to close the gap of pre-training.
4. The performance gain of co-tuning compared with using a tuning teacher is hard to understand for me. In both cases, there are gaps between the initial student and teacher models. What is the in-depth reason that we should use co-tuning?
5. It is unclear whether the scale and shift operation is spatial-wise. If not, this can be merged with the linear projector, or can be replaced by more powerful tools like LoRA. If yes, some visualization would be helpful to illustrate what they learn.
6. How about making the feature projector non-linear?
7. It seems that the teacher is also affected by the distillation loss. How about detaching the distillation loss $L_{KD}$ and/or $L_{MSE}$ from the teacher?
8. The experiments are conducted on relatively small datasets. Larger datasets like DomainNet are recommended. Also, since this is a multimedia conference, a wider range of application is encouraged like tuning the generative model, e.g., DreamBooth.

**Suitability:**

2

---

### Official Review · Reviewer_d68A · 2024-06-04

**Rating:** 5
**Confidence:** 2

**Summary:**

This work proposed a co-tuning knowledge distillation framework that trains the large teacher model and the small student model in a collaborative scheme. The authors proposed to add adapter modules to the large model to be co-tuned with the student model. A projection module is applied to map the features generated by the small model to the hidden space of the larger model, so that a matching loss can be calculated in the later distillation phase. Then the large-small models are trained simultaneously, which is the "collaborative distillation phase". Empirical results showed very impressive perforamnce of the proposed method when compared with existing knowledge distillation methods.

**Strengths:**

The authors conducted intensive empirical experiments and showed very impressive performance achieved by the proposed method. The authors also did carefully designed ablation study to show the impact of feature collaborative knowledge distillation in Section 5.3, which is very informative to deconstruct the effects brought by different components.

**Limitations:**

Overall, I did not grasp the insight of why co-tuning the teacher model (the adapter modules) is helpful with improving generalization of the student model.

The writing of the manuscript sometimes feels repetitive and redundant, and is unclear in many places.
1. The beginning of Sec 3.2 is repeating the contents described in the following subsections.
2. The definition of $k$ repeats twice around eqn (4).
3. It is not clear what FCKD in Table 3 actually refers to. If it is the collaborative training scheme, then what does a "static" or "trainalbe" teacher mean? Is it something different from FCKD?

**Suitability:**

3

---

### Meta-Review · Area_Chair_83nB · 2024-07-04

**Recommendation:** Accept (Poster)
**Confidence:** 4

**Metareview:**

The manuscript has been reviewed by four reviewers, among whom three recommended acceptance, indicating that the majority agrees that the submission meets the bar of MM.

The AC agrees with the consensus; considering the merits of the manuscript, the AC believes the manuscript deserves to be presented to a large audience.

Please, however, do account for the comments in the final version. Congrats!